# Effects of Green Tea Catechins on Prostate Cancer Chemoprevention: The Role of the Gut Microbiome

**DOI:** 10.3390/cancers14163988

**Published:** 2022-08-18

**Authors:** Nagi B. Kumar, Stephanie Hogue, Julio Pow-Sang, Michael Poch, Brandon J. Manley, Roger Li, Jasreman Dhillon, Alice Yu, Doratha A. Byrd

**Affiliations:** 1Cancer Epidemiology Program, Moffitt Cancer Center and Research Institute, Tampa, FL 33612, USA; 2Genitourinary Oncology, Moffitt Cancer Center and Research Institute, Tampa, FL 33612, USA; 3Anatomic Pathology, Moffitt Cancer Center and Research Institute, Tampa, FL 33612, USA; 4Gastrointestinal Oncology, Moffitt Cancer Center and Research Institute, Tampa, FL 33612, USA

**Keywords:** prostate cancer, green tea catechins, microbiome, chemoprevention

## Abstract

**Simple Summary:**

Green tea is known for its health benefits deriving from molecules called green tea catechins (GTCs). GTCs have been demonstrated to influence molecular pathways to halt the progression of prostate cancer (PCa) and may be of particular benefit to men with low-risk PCa who are placed on active surveillance. Administering GTCs may provide patients an opportunity to be actively engaged in their treatment and help prevent cancer progression. Importantly, the trillions of microbes in the gut (the gut microbiome) metabolize GTCs, making them more accessible to the body to exert their health effects. Additionally, the gut microbiome influences multiple other processes likely involved in PCa progression, including regulating inflammation, hormones, and other known/unknown pathways. In this review, we discuss (1) the role of GTCs in preventing PCa progression; (2) current evidence for associations of the microbiome with PCa; and (3) utilizing the microbiome to identify markers that may predict improved response to GTCs to enhance clinical decision-making.

**Abstract:**

Accumulating evidence supports green tea catechins (GTCs) in chemoprevention for prostate cancer (PCa), a leading cause of cancer morbidity and mortality among men. GTCs include (−)-epigallocatechin-3-gallate, which may modulate the molecular pathways implicated in prostate carcinogenesis. Prior studies of GTCs suggested that they are bioavailable, safe, and effective for modulating clinical and biological markers implicated in prostate carcinogenesis. GTCs may be of particular benefit to those with low-grade PCas typically managed with careful monitoring via active surveillance (AS). Though AS is recommended, it has limitations including potential under-grading, variations in eligibility, and anxiety reported by men while on AS. Secondary chemoprevention of low-grade PCas using GTCs may help address these limitations. When administrated orally, the gut microbiome enzymatically transforms GTC structure, altering its bioavailability, bioactivity, and toxicity. In addition to xenobiotic metabolism, the gut microbiome has multiple other physiological effects potentially involved in PCa progression, including regulating inflammation, hormones, and other known/unknown pathways. Therefore, it is important to consider not only the independent roles of GTCs and the gut microbiome in the context of PCa chemoprevention, but how gut microbes may relate to individual responses to GTCs, which, in turn, can enhance clinical decision-making.

## 1. Introduction

The American Cancer Society estimates that there will be 268,490 and 34,500 new cases of and deaths due to prostate cancer (PCa) in the United States (US) in 2022, respectively [1]. Over the past two decades, PCa screening via serum prostate-specific antigen (PSA) led to substantial increases in detection of low-risk PCas (Gleason score ≤ 6), which pose little risk of either metastatic spread or death [2,3,4,5]. Conversely, over-treatment is a well-documented consequence of over-detection of PCa, predominantly occurring among men with low-risk PCa who may be subject to multiple treatment-related morbidities with negligible or no benefit towards cancer-specific survival [4,6]. Thus, the recommended guideline for the management of low-risk disease is active surveillance (AS). However, there are several identified challenges with AS, ranging from concerns with under-grading [7,8,9,10,11,12,13], patient-related factors (e.g., anxiety, depression, doubts about the possible progression of disease), and higher decisional conflict regarding the selection of AS [14,15,16], leading many to ultimately opt for a treatment that does not beneficially change tumor characteristics. On the other hand, men on AS are a highly motivated subgroup eager to make positive lifestyle changes to reduce their risk of PCa progression [16,17,18,19,20,21], providing an optimal opportunity to intervene during this window with promising chemopreventive agents for PCa.

Previous strategies for PCa chemoprevention included 5-alpha-reductase inhibitors, finasteride, dutasteride [22,23,24], trace element selenomethionine, and/or vitamin E. Collectively, these agents demonstrated greater risk for high grade disease [25] or no reduction in risk of PCa progression in large phase III trials, severely limiting their clinical adoption [23]. To date, there is minimal evidence available for the efficacy of any one agent or strategy for chemoprevention of PCa among men on AS. Therefore, the goal of our team for PCa chemoprevention is to utilize a systematic, broad-spectrum approach [26] that involves an agent shown to (a) be bioavailable; (b) have an excellent safety profile; (c) produce robust targeting of multiple relevant molecular pathways; and (d) modulate measurable intermediate endpoint biomarkers correlated with early clinical progression of PCa—an approach that collectively may be more effective than agents evaluated to date. Our team and others have evaluated several approaches (i.e., diet interventions) and agents (selenium, vitamin E, isoflavones, lycopene n-3 fatty acids, and green tea catechins, or GTCs) targeting prostate carcinogenesis.

Human PCa is a complex heterogeneous disease. The central driving forces of prostate carcinogenesis include acquisitions of diverse sets of hallmark capabilities, aberrant functioning of androgen receptor signaling, deregulation of vital cell physiological processes, inactivation of tumor-suppressive activity, and disruption of prostate gland-specific cellular homeostasis. Thus, the molecular complexity and redundancy of oncoprotein signaling in PCa demands for concurrent inhibition of multiple hallmark-associated pathways [27]. The ultimate goal for clinical cancer chemoprevention is to utilize a systematic, broad-spectrum approach that involves identifying and evaluating agents that can: (a) produce robust and concurrent inhibition of multiple hallmark-associated pathways in the target tissue/microenvironment; (b) address the underlying biology of carcinogenesis; and (c) enhance bioavailability and half-life with minimal toxicity in exceptionally high-risk populations [26,28]. GTCs comprise (−)-epigallocatechin-3-gallate (EGCG), (−)-epicatechin, (−)-epigallocatechin (EGC), and (−)-epicatechin-3-gallate. Among the agents evaluated to date, EGCG in particular has been demonstrated to affect molecular pathways implicated in prostate carcinogenesis.

The objective of this review is to summarize the current research on the safety and effectiveness of GTCs in modulating prostate carcinogenesis based on population, in vitro, pre-clinical and early clinical trials. Although previous reviews have examined the pre-clinical and early phase trials of GTCs and PCa [29,30,31,32,33], our review will additionally identify discrepancies in the results of previous studies and examine the early and evolving data on the role of the gut microbiome in modulating the bioavailability, safety, and anticarcinogenic properties of GTCs in prostate carcinogenesis.

## 2. GTCs: Promising Agent for PCa Chemoprevention

The most abundant constituents of green tea are the polyphenols, which are catechins that represent 30–40% of the dry weight of the tea leaves. The catechins in green tea belong to the flavon-3-ols of the polyphenol family [34]. Laboratory studies have identified EGCG as the most potent modulator of molecular pathways thought to be relevant to prostate carcinogenesis [35,36,37,38]. In the past two decades, research studies have shown that GTCs influence multiple biochemical and molecular cascades that inhibit several hallmarks of carcinogenesis relevant to prostate carcinogenesis. With an acceptable safety profile, GTCs are ideal candidates for PCa chemoprevention. Laboratory studies demonstrate that EGCG can affect several cancer-related proteins, including p27, Bcl-2 or Bcr-Abl oncoproteins, Bax, matrix metalloproteinases (MMP-2 and MMP-9), the androgen receptor (particularly important in PCa development and progression), epidermal growth factor receptor, activator protein 1, and some cell cycle regulators [29,35]. Using cell culture systems, Adhami et al. [39] were able to show that EGCG induces apoptosis, cyclin kinase inhibitor WAF-1/p21-mediated cell cycle-dysregulation, and cell growth inhibition. In cDNA microarrays, EGCG treatment of LNCaP cells induced genes that exhibit growth-inhibitory effects and repressed genes belonging to the G-protein signaling network [40]. The ubiquitin/proteasome pathway plays a critical role in activation of the cellular apoptotic program and the regulation of apoptosis [41]. Our work demonstrated that GTC specifically inhibits the chymotrypsin-like activity of the proteasome in several tumor and transformed cell lines, including prostate cell lines, resulting in the accumulation of two natural proteasome substrates–p27 (Kip1) and nuclear factor kappa B (NF-ĸB) inhibitor alpha, which inhibit transcription factor NF-ĸB, leading to growth arrest in the G(1) phase of the cell cycle. Synthetic analogs of EGCG were observed to be more potent as proteasome inhibitors compared to EGCG. Polyphenon E^®^ (Poly E) and Sunphenon^®^ 90D are standardized formulations of green tea containing 50% of the catechins from EGCG. We observed that Poly E^®^ (>50% EGCG, 80% total catechins) preferentially inhibits the proteasomal chymotrypsin-like activities over other activities, with an IC50 value of 0.88 µM [41,42,43,44]. Standardized GTC formulations of Poly E^®^ and Sunphenon^®^ 90D in equal concentrations were evaluated in vitro. Pre-treatment with Sunphenon^®^ 90D downregulated NF-ĸB in H_2_O_2_-treated C2C12 cells, while activating caspase-3 (Figure 1) [45]. Incubation of human primary osteoblasts with Sunphenon^®^ 90D significantly reduced oxidative stress and improved cell viability [46]. EGCG has been shown to have both anti-inflammatory properties, such as through the influence of T-cell proliferation and inhibition of NF-ĸB, and neuroprotective properties by acting as a free radical scavenger [47,48]. More specifically, EGCG’s antioxidant properties deplete reactive oxygen species, thus preventing DNA damage and inhibiting NF-ĸB-induced inflammation, angiogenesis, and cell survival that could otherwise propel cancer development and progression [49]. In summary, we and others have reported convincing evidence suggesting that GTCs inhibit proliferation and cell cycle events and induce apoptosis through multiple mechanisms.

The association of green tea intake with PCa risk has been investigated in several epidemiological studies. In a meta-analysis of 9 case-control studies, there was a statistically significant 57% lower risk of PCa, comparing subjects with the highest relative to lowest green tea consumption, whereas there was a null association in a meta-analysis of 4 cohort studies [50]. Similar results were observed in a more recent meta-analysis of 3 case-control and 4 cohort studies: no statistically significant associations were observed across cohort studies, while a statistically significant 55% lower odds of PCa was observed for highest versus lowest green tea intake in the case-control studies [51]. This inconsistency could be due, in part, to differences in study design, residual confounding factors such as by diet/lifestyle and biological factors, and varying formulations and subtypes of green tea studied. These studies were mostly limited to men in Asian countries, where approximately 20% of green tea is consumed globally and where mortality from PCa is the lowest compared to Western populations [36], where green tea consumption is a more recent phenomenon. Asian men who migrate to the US have a relatively increased risk of PCa compared to their counterparts in their countries of origin, potentially as a result of acculturation and adoption of Western diets [38]. Although the above study findings have been mixed—potentially due to confounding by variation in geographical location, tobacco and alcohol use, and other lifestyle factors (mainly diets) [37,52]—taken together, studies among Asian populations demonstrate a protective effect of GTCs as related to PCa [37,38,53]. Another highly plausible confounder of GTC-PCa associations is the gut microbiome, which has increasingly been implicated in the modulation of carcinogenesis. The gut microbiome comprises densely populated commensal and symbiotic microbes [54] whose composition is highly influenced by the host’s dietary intake. The gut microbiome also produces metabolically active metabolites that interact with host-signaling pathways and gene expression, impacting cancer initiation and progression [55,56]. Multiple studies have observed differences in the gut microbiome between various racial and ethnic groups, even amongst those living in the same community. These differences are potentially attributed to lifestyle, dietary, social, and other uncharacterized exposures that result in variations across racial and ethnic groups [57,58]. Using fecal shotgun metagenomic data analyzed amongst 106 Japanese individuals compared with those of 11 other nations, the composition of the Japanese gut microbiome was more abundant in the phylum Actinobacteria, in particular, genus *Bifidobacterium*, compared to others [59]. In line with increased PCa rates in Asian populations living in the US, studies have shown that the gut microbiome of Southeast Asian immigrants changes after migration to the US [60], potentially indicative of an incompatibility between the incorporation of Western lifestyles with the traditionally harbored microbiome of this population [61]. These studies have provided the basis for understanding that the gut microbiome can act as an important mediating factor in investigations of diet and lifestyle differences that potentially promote cancer risk.

## 3. Pre-Clinical Evidence of the Safety and Effectiveness of GTCs in PCa Carcinogenesis

Several promising pre-clinical studies of GTC effects on prostate carcinogenesis were completed that were highly clinically relevant [39,62,63,64,65,66]. In studies evaluating oral GTCs (vs. pure EGCG) administered to transgenic adenocarcinoma of the mouse prostate (TRAMP) mice, greater bioavailability [29,35,62,63,64,65,66] of GTC was observed compared to administering EGCG alone [64]. Oral infusion of a polyphenolic fraction isolated from green tea extract at a human achievable dose (i.e., six cups of green tea per day) in a TRAMP mouse model, compared to water-fed mice [62], demonstrated significant delays in primary prostate tumor incidence and burden. Overall, they observed a decrease in prostate (64%) and genitourinary (72%) weight from baseline weight, inhibition of serum insulin-like growth factor-1 (IGF-1), and restoration of insulin-like growth factor binding protein-3 levels (IGFBP-3). Additionally, a significant reduction in the protein expression of proliferating cell nuclear antigen and apoptosis in the prostate was observed in GTC-fed mice compared to water-fed mice, resulting in reduced dissemination of cancer cells, thereby causing inhibition of development, progression, and metastasis to distant organ sites. Our team evaluated [65] the safety and efficacy of GTCs at various doses (200, 500, and 1000 mg EGCG in GTC/kg/day) in reducing the progression of PCa in a TRAMP mouse model. Significant decreases in the number and size of tumors in treated TRAMP mice were observed compared with untreated animals. We observed a dose-dependent inhibition of metastasis in GTC-treated mice (*p* = 0.0003). After 32 weeks of treatment with standardized formulation of GTC, it was found to be well-tolerated with no evidence of toxicity in C57BL/6J mice [65]. Apart from significant reductions in tumor size and multiplicity, GTCs also prevented metastatic progression of PCa in the TRAMP and other relevant mouse models. Collectively, these findings from pre-clinical studies, using doses relevant for translation to human clinical trials, provide evidence for safety and chemopreventive effects of GTCs.

## 4. Clinical Evidence of Bioavailability, Safety, and Effectiveness of GTCs in Modulating Prostate Carcinogenesis

Early phase I/II studies [67,68,69,70,71,72,73,74,75,76] conducted over the past decade found that doses of GTCs containing 200–1200 mg of EGCG per day (Poly E®) were tolerated by subjects, including men with precancerous lesions such as high-grade prostatic intraepithelial neoplasia, atypical small acinar proliferation, or early stage PCas.

Additionally, early phase I trials assessing standardized formulations of GTCs (Sunphenon 90D^®^) demonstrated increasing doses of plasma EGCG with increasing doses of the supplement [77,78,79]. Oral intake of GTCs in healthy subjects containing 225, 375, and 525 mg EGCG (Sunphenon^®^ 90D) demonstrated a significant dose-dependent increase in plasma concentrations of EGCG to 657, 4300, and 4410 pmol EGCG/mL, respectively [80]. Consumption of Sunphenon^®^ 90D containing 246 mg EGCG significantly increased plasma EGCG, which was highly correlated with attenuation of plasma phosphatidylcholine hydroperoxide levels, a marker of antioxidant capacity. Although increased bioavailability (as indicated by higher concentrations of EGCG in plasma) [81,82,83] occurs when GTCs are consumed in a fasting state [69] as opposed to a fed state, increased toxicity has also been reported when GTCs are taken in a fasting state. Similarly, increased bioavailability and tolerance to a multiple dosing schedule compared to a single daily dose of EGCG has been reported in phase II trials [67,75,84]. A summary of the concentration of GTCs in plasma with intervention trials targeting men at high risk for PCa is presented in Table 1. Mean plasma concentrations of EGCG varied among all these trials, potentially due to varying duration of intervention, doses, methods used in analyzing plasma EGCG, ethnicity of the target population, and nutritional and lifestyle habits. Another potential explanation may be differences in gut microbial capacity to process GTCs, as described below.

Overall, prior studies support that GTCs are generally safe for consumption in human populations. A phase II/III trial (NCT00799890) was recently completed to evaluate the effect of 200–800 mg Sunphenon^®^ 90D on attenuating brain atrophy, targeting patients with primary or secondary chronic-progressive multiple sclerosis treated for 36 months. An additional study (NCT00951834), using a maximum dose of 800 mg EGCG for 18 months to target early Alzheimer’s, has been completed with results pending. No toxicities have been reported in either of these trials. In a more recently reported phase II clinical trial that evaluated the effects of 1315 mg of total catechins, containing 843 mg of EGCG, vs. placebo in modulating mammographic density, 1075 women were evaluated in a 12-month intervention. Overall, 26 women (5.1%) in the green tea extract arm developed moderate to severe abnormalities in liver function tests during the intervention period [86,87]. In three randomized trials [84,88,89,90] of the effects of GTCs (800 mg EGCG), or green tea as a beverage, in men diagnosed with localized PCa prior to prostatectomy, no toxicities were observed. These trials did not collect and analyze samples to assess interactions of the gut microbiome with GTC safety and toxicity.

A summary of the changes observed in intermediate endpoint biomarkers of PCa among Phase II GTC clinical trials is presented in Table 2. The findings from our study [75] and those of Bettuzzi et al. [67,68] suggest that a daily intake of the standardized GTC formulation administered non-fasting for 12 months in divided doses: (a) accumulates in plasma; (b) reduces serum PSA; and (c) reduces the cumulative rate of progression to PCa with no toxicities [67,68,75]. In the study of the effects of green tea beverages [88], nuclear staining of NF-κB was significantly decreased in radical prostatectomy (RP) tissue of men consuming GTC (*p* = 0.013), but not black tea (*p* = 0.931), compared to water control. Further, GTCs were detected in prostate tissue from 32 of 34 men consuming green tea but not in the other groups; evidence of a systemic antioxidant effect was observed (i.e., reduced urinary 8-hyroxydeoxy-guanosine) only with green tea consumption (*p* = 0.03) [88]. In randomized trials of the effects of GTCs (800 mg EGCG) or green tea as a beverage among men diagnosed with localized PCa, prior to prostatectomy, a reduction in serum PSA was observed [88,89]. Nguyen et al. [84] observed that the proportion of subjects who had a decrease in Gleason score between biopsy and surgical specimens was greater among those randomized to GTCs, but this finding was not statistically significant for the full duration of the intervention. In an open-label, single-arm, two-stage phase II clinical trial, 26 men with positive prostate biopsies received 800 mg EGCG/day (Poly E®) for 3–6 weeks until undergoing RP. EGCG administration lowered serum concentrations of hepatocyte growth factor (HGF), vascular endothelial growth factor, IGFBP-3, IGF-1, and PSA in these patients, with no elevation of liver enzymes [89]. More recently, Lane et al. [85] completed a 6-month randomized controlled trial of green tea and lycopene among men with elevated serum PSA but negative prostate biopsies. They randomized men to consume food sources of each of these agents, to standardized formulations, or to placebo. Plasma levels of both lycopene and EGCG were higher in the treatment arms compared to the placebo arm with concentrations among the dietary source arm of these formulations being greater than the capsule group and the placebo arm. All interventions were tolerated well by the participants; however, men preferred the capsules to using food sources of lycopene and green tea. No biomarkers of disease progression were assessed in this study.

In summary, evidence from epidemiological, in vitro, pre-clinical, and early phase trials completed by our team and others have shown that the standard GTC formulations (a) accumulate in plasma and tissue; (b) reduce serum PSA and cumulative rate of progression to PCa; and (c) are potent inhibitors of PCa carcinogenesis through multiple mechanisms without toxicities at these doses, establishing the evidence needed for further development of GTCs in phase II clinical trials targeting men at exceptional risk or those diagnosed with low risk PCas. Self-reported patient race/ethnicity, medical history, family history of cancer, lifestyle factors such as smoking and alcohol use, and dietary intake have been accounted for in many of these studies; however, to date, the role of the gut microbiome in the absorption, safety, and modulation of prostate carcinogenesis has not been evaluated. Although the data on the safety, effectiveness, and potential mechanism of GTCs in prostate carcinogenesis appears promising, there are gaps in knowledge pertaining the role of the gut microbiome in modulating the bioavailability and toxicity of GTCs and in prostate carcinogenesis. With significant variations observed in GTC bioavailability (Table 1) as well as in modulation of intermediate endpoint biomarkers with GTCs (Table 2) in prior clinical trials, it is imperative to evaluate the contribution of the gut microbiome to modulating the interrelationships among GTC chemoprevention and PCa progression.

## 5. The Gut Microbiome, PCa, and GTCs

Predictive biomarkers of responses to secondary chemoprevention are presently lacking. Identification of biomarkers, such as the gut microbiome, predictive of favorable clinical responses to secondary chemoprevention has the potential to substantially facilitate clinical decision-making. Numerous studies found that the gut microbiome directly effects drug metabolism, efficacy, and toxicity, potentially affecting disease development and progression [91]. For example, in oncology, there exists convincing evidence to support that the antitumor effects of immunotherapies can be enhanced or inhibited by the gut microbiome [92,93].

The gut microbiome likely has critical roles in regulating the bioavailability of GTCs and absorption of bioactive phenolic GTC metabolites, as demonstrated in laboratory and pre-clinical models (Figure 2). Although dietary polyphenols are absorbed by the small intestine, accumulating evidence suggests that they are metabolized to a greater extent in the colon by bacterial enzymes [94,95]. EGCG is hydrolyzed by bacteria to gallic acid or EGC and further converted to multiple metabolites, such as 5-(3,5-dihydroxyphenyl)-4-hydroxyvaleric acid and 5-(3′,5′-dihydroxyphenyl)-g-valerolactone [95]. These metabolites are then either taken up via the portal vein and transported to the liver or excreted in the feces.

On the other hand, several pharmacologic agents, including GTCs, were shown to influence gut microbiome composition and function. For example, in a study of 10 volunteers who drank 1000 mL of green tea daily for 10 days, Bifidobacteria abundance was increased [101]. In multiple animal studies, green tea polyphenols had similar effects on Bifidobacteria and other effects, including decreasing the Firmicutes/Bacteroidetes ratio [102,103]. In turn, gut microbiome composition and function may directly and indirectly influence PCa progression, such as through production of metabolically active metabolites or regulation of hormones and inflammation, as described below [104,105,106]. Given the substantial preliminary evidence for interrelationships among GTCs, the gut microbiome, and prostate carcinogenesis, it is highly likely that the gut microbiome may mediate etiological effects of GTCs, including effects on PCa progression and development of adverse events; however, little is known regarding these interrelationships among humans.

The gut microbiome has biologically plausible roles in PCa such as via its influence on hormone and inflammation regulation and production of metabolically active metabolites [104,105,106]. For example, gut microbes produce sex hormones, such as androgen, and, in a study by Pernigoni et al., multiple species among mice and humans produced androgens from androgen precursors, in turn promoting progression of castrate resistant PCa [107]. In a study of both mice and PCa patients, Proteobacteria was increased after antibiotic exposure, and was in turn associated with development of PCa in mice and with metastasis of PCa among humans [108]. In a study of mice on a high-fat diet, the resultant alterations to the mice fecal microbiome promoted histamine biosynthesis and increased inflammatory cancer cell growth [109]. Previous human studies of the gut microbiome and PCa included a case-control study comparing 16S rRNA sequenced fecal bacteria among 64 men with PCa and 41 without PCa, finding differences in beta diversity, higher abundances of Bacteroides and Streptococcus species, and differences in folate and arginine pathways [110]. Another case-control study compared the gut metagenome among 8 men with benign prostatic conditions and 12 men with intermediate or high risk clinically localized PCa, finding higher relative abundance of Bacteriodes massiliensis and lower relative abundances of Faecalibacterium prausnitzii and Eubacterium rectalie amongst men with intermediate/high-risk PCa [111]. In a comparison of men with and without prostate enlargement, the ratio of Firmicutes to Bacteroidetes was higher among men with enlarged prostates, potentially related to prostate inflammation [112]. Finally, evidence supports the study of the gut microbiome across the disease continuum of PCa, with evidence demonstrating that the gut microbiome may be modified by PCa treatment, including androgen deprivation therapy, among more advanced PCa patients [113].

## 6. Challenges and Future Directions

There is currently sufficient evidence that establishes the need to evaluate the role of the gut microbiome in modifying the response to GTCs among men diagnosed with PCa. However, there are several challenges and pitfalls pertaining to studying GTCs in the context of modification by the gut microbiome. At present, biospecimens and data available to study the role of the gut microbiome in the effects of GTCs are sparse and further research is clearly needed among diverse populations. Studying the gut microbiome itself presents several challenges, as it is a complex, dynamic ecosystem that is driven by numerous known and unknown factors, such as dietary intake, requiring comprehensive measurement of potential confounding factors. It has been well documented that methods for stool collection, DNA extraction, and sequencing can influence downstream gut microbiome metrics, potentially resulting in inconsistent study findings that hinder progress in the field. To ensure high-quality, reproducible results, it is critical to establish contemporary and validated methodologies and to optimize protocols and procedures for fecal sampling, handling, processing, and microbiome analyses [114,115].

To fill existing gaps in knowledge, detailed characterizations of the gut microbiome and its metabolites among extensively phenotyped human subjects are needed. Although sequencing of the 16S rRNA gene classifies bacteria based on conserved single marker genes, there is a lack of detailed resolution. Shotgun metagenomic sequencing, which comprises the untargeted sequencing of all DNA present in a sample, provides more detailed taxonomic information than 16S rRNA sequencing. Characterizing microbial genes, strains, and functions may provide deeper insight into GTC-gut microbiome interactions. In addition, the gut microbiome and metabolome have moderate-to-high intraindividual variability and are ‘high-dimensional’ in that there are typically large numbers of microbes/metabolites relative to the numbers of subjects. As a result, collecting repeat samples is particularly useful for reducing bias in estimating effects/associations and increasing statistical power [116]. Other challenges include addressing limitations of previous human studies of microbiome-PCa associations, including small sample sizes, inclusion of more advanced PCas, and cross-sectional design leading to concerns with reverse causality. Studies among large populations of men with serial microbiome and intermediate biomarker endpoint assessments are critically needed.

Future studies evaluating GTCs in prostate carcinogenesis may also include a metabolomics approach to assess EGCG- and microbiome-related metabolites from stool samples pre- and post-treatment with GTCs. In addition, these studies must include the evaluation of the correlation among specific microbial species/strains with (a) plasma levels of EGCG; (b) multiple markers of toxicity and safety; and (c) surrogate endpoint biomarkers, such as serum PSA, as an indicator for PCa progression, to provide timely evidence for a role of the gut microbiome in mediating the effects of GTCs on PCa progression. To our knowledge, studies collecting serial stool samples longitudinally to measure the microbiome in relation to these intermediate biomarkers of prostate carcinogenesis are currently unprecedented. Further, as in all biomedical research, there should be an emphasis in recruiting and studying disproportionately affected populations. In PCa, African American (AA) men are known to have the highest PCa risk. There is accumulating evidence that exposures associated with race may collectively and individually influence gut microbiome composition [57,58]. Therefore, with the inclusion of AA men in these clinical trials, we may be able to provide data to inform secondary chemoprevention efforts among this particularly at-risk population by studying a comprehensive biomarker (the gut microbiome). Finally, this review is focused and specific to the current data on the safety, effectiveness, and molecular mechanisms of GTCs in prostate carcinogenesis. Other cancers, like breast and colon cancers, may similarly be impacted by both GTCs and the gut microbiome. Collectively, these studies are critical in understanding the dynamics of the gut microbiome as we develop and evaluate promising agents such as GTCs for cancer chemoprevention.

## Figures and Tables

**Figure 1 cancers-14-03988-f001:**
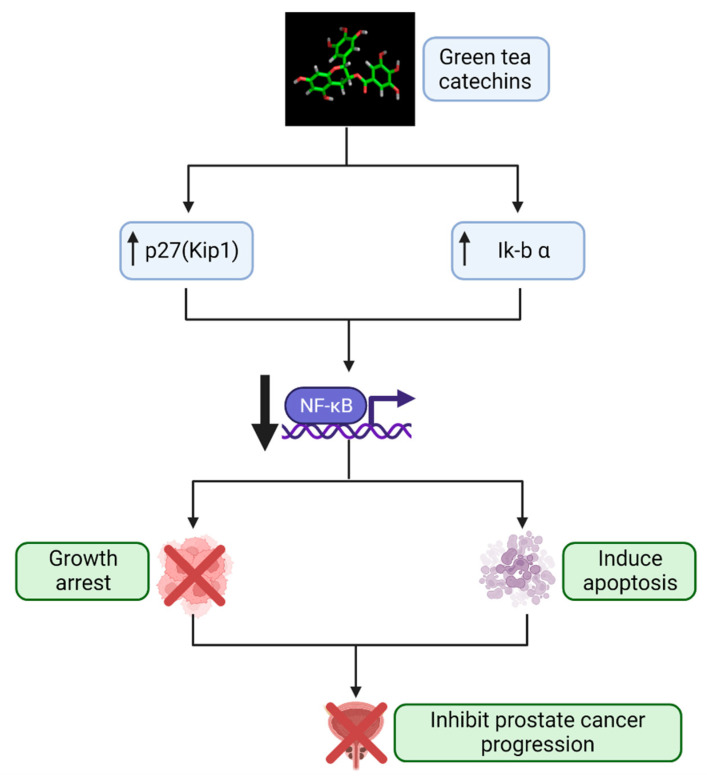
Mechanistic pathway by which GTCs prevent PCa progression. In vitro studies [41,42,43,44] demonstrate that GTCs block proteasomal activity in PCa cells, leading to build-up of proteasomal substrates Kip1 and Ik-b α that subsequently downregulate the activity of NF-κB. This inhibits the cell cycle and elicits apoptosis in these PCa cells. GTCs, green tea catechins; Ik-b α, NF-κB inhibitor alpha; NF-κB, nuclear factor kappa B; PCa, prostate cancer. Created with Biorender.com (accessed on 1 July 2022).

**Figure 2 cancers-14-03988-f002:**
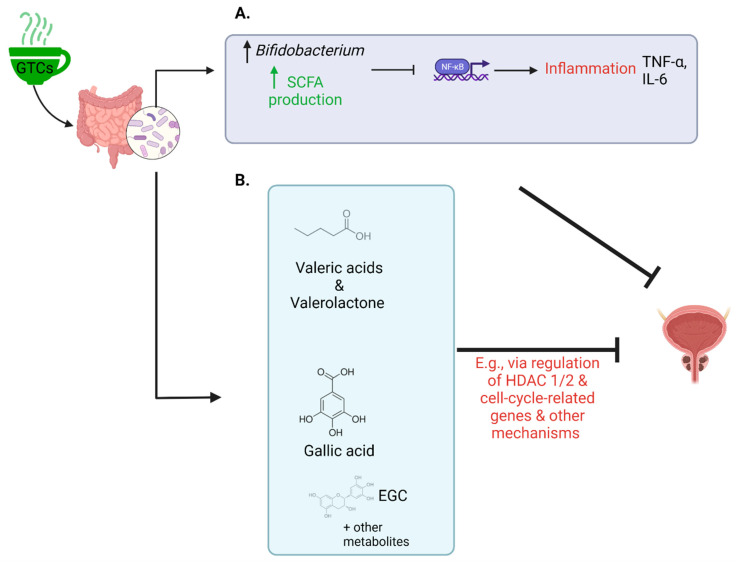
Examples of chemopreventive effects of GTCs in the context of PCa via gut microbiome modulation. (**A**) GTCs like EGCG have been evidenced to alter microbial composition, such as increasing abundance of Bifidobacterium [95]. This genus, for example, is known to increase production of SCFAs [95,96] which inhibit inflammatory pathways initiated by NF-κB that would otherwise propel carcinogenesis [97]. (**B**) The gut microbiome can enzymatically alter GTCs like EGCG to produce metabolites including gallic acid, EGC, valeric acid, and valerolactone, that subsequently travel to the bloodstream to exert potential chemopreventive benefits (e.g., regulating HDAC 1 and 2 and suppressing cell-cycle-related genes) [95,98,99,100]. EGC, epigallocatechin; EGCG, epigallocatechin gallate; GTCs, green tea catechins; HDAC, histone deacetylase; IL-6, interleukin-6; NF-κB, nuclear factor kappa B; SCFA, short chain fatty acid; TNF-α, tumor necrosis factor alpha. Created with Biorender.com (accessed on 1 July 2022).

**Table 1 cancers-14-03988-t001:** Concentration of GTCs in plasma in interventional trials targeting men with PCa.

Author; Target Population	Dose of EGCG (mg)	Duration of Intervention	Plasma EGCG Concentration after Intervention
Nguyen et al. [84]; PCa patients prior to RP	800 (Poly E®)	3–6 weeks	146.6 pmol/mL
Kumar et al. [75]; Men with HGPIN	200 (BID) (Poly E®)	1 year	12.3 ng/mL (SD, 24.8) fed
Bettuzi et al. [67]; Men with HGPIN	200 (TID)	1 year	NA
Lane et al. [85]; Men with elevated PSA or negative prostate biopsy for PCa	GTC drink GTC capsules	6 months	24.9 nmoL/L 12.3 nmoL/L

BID, twice a day; EGCG, (−)-epigallocatechin-3-gallate; GTC, green tea catechins; HGPIN, high-grade prostatic intraepithelial neoplasia; PCa, prostate cancer; Poly E, Polyphenon E; PSA, prostate specific antigen; RP, radical prostatectomy; SD, standard deviation; TID, three times a day.

**Table 2 cancers-14-03988-t002:** Changes in intermediate endpoint biomarkers of PCa observed in Phase II clinical trials using GTCs.

Target Population (Ref)	Number of Subjects	Dose of GTC (EGCG)	Duration of Intervention	Biomarkers Observed
HGPIN (Betuzzi et al. [67,68])	60	200 mg TID	12 months	Reduction in progression to PCa in treatment arm Improvement in prostate symptom score
HGPIN (Kumar et al. [75])	97	200 mg BID Poly E®	12 months	Cumulative rate of PCa plus ASAP among men with HGPIN without ASAP at baseline, revealed a decrease in this composite endpoint: (*p* < 0.024). Decrease in ASAP diagnoses on the Poly E® (0/26) compared with the placebo arm (5/25). Decrease in serum PSA was observed in the Poly E arm [−0.87 ng/mL; 95% CI, −1.66 to −0.09].
PCa patients (Henning et al. [88])	113	6 cups of green tea, black tea or water	3–8 weeks	Nuclear staining of NF-κB was significantly decreased in RP tissue of men consuming green tea (*p* = 0.013) but not black tea (*p* = 0.931) compared to water control. Tea polyphenols were detected in prostate tissue from 32 of 34 men consuming green tea but not in the other groups. Evidence of a systemic antioxidant effect was observed (reduced urinary 8OHdG) only with GTC consumption (*p* = 0.03). Significant decrease in serum PSA levels (*p* < 0.05).
PCa patients (McLarty et al. [89])	26	800 mg of EGCG Poly E®	3–6 weeks	Significant reduction in serum levels of PSA, HGF, and VEGF in men with PCa after brief treatment with EGCG (Poly E®), with no elevation of liver enzymes.
PCa patients-pre-prostatectomy (Nguyen et al. [84])	52	800 mg of EGCG Poly E®	3–6 weeks	Proportion of subjects who had a decrease in Gleason score between biopsy and surgical specimens was greater in those on Poly E® but was not statistically significant. Favorable but not statistically significant changes in serum PSA, serum insulin-like growth factor axis, and oxidative DNA damage in blood leukocytes.

Abbreviations: 8OHdG, 8-hydroxydeoxy-guanosine; ASAP, atypical small acinar proliferation; BID, twice a day; CI, confidence interval; GTC, green tea catechins; HGF, hepatocyte growth factor; HGPIN, high-grade prostatic intraepithelial neoplasia; EGCG, epigallocatechin-3-gallate; PCa, prostate cancer; Poly E, polyphenon E; PSA, prostate specific antigen; RP, radical prostatectomy; TID, three times a day; VEGF, vascular endothelial growth factor.

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
