# Peer review of "Effects of Green Tea Catechins on Prostate Cancer Chemoprevention: The Role of the Gut Microbiome"

_cancers, 2022, doi:10.3390/cancers14163988_

Round 1

Reviewer 1 Report

This is a comprehensive review on the subject of prostate cancer chemoprevention by green tea catechins in the new light of the role of gut microbiome from Dr. Kumar's group who is leader in the field. The information covered in this article would provide critical information to the researchers in this important area of research.

Reviewer 2 Report

The overall manuscript is well written and covers an interesting area and would be helpful for readers. However there are certain flaws that need to be addressed before it can be accepted for publication.

1.      The article lacks figures and the authors are advised to add figures as it is necessary for an article to become attractive and aid in understanding for a broader audience.

2.      There are many grammatical errors throughout the manuscript and also several long sentences are there that need to restructured.

3.        Check for consistent usage of abbreviations throughout and avoid using abbreviations at first hand.

4.      Adding latest article pertaining to ECGC will aid in improvement. https://doi.org/10.1007/978-981-16-4558-7_7

5.      Tables need formatting in accordance with journal policies.

6.      A summary as written in text shall be depicted in a graphical abstract form.

Reviewer 3 Report

This is a well-conceived and timely review of the potential utility and future research directions for Green Tea polyphenols as an effective chemopreventative for early-stage prostate cancer. There has been a large body of literature on preclinical studies of Green Tea Catechins  (GTCs) but very few studies on the potential role of microbiota on the anti-cancer effects of GTCs. The authors have a well-developed rationale to direct readers on the pitfalls of previous studies or multiple studies with the same objective but lack uniformity in reporting, for example, in studies on the GTCs role as chemopreventives in Active Surveillance patients of prostate cancer, PSA progression was not reported, similarly, prostate tumor-specific tests were not undertaken in year-long volunteer based GTC consumptions. The authors have lucidly explained these critical details and the overall quality of the manuscript is very high, or at least in this reviewer's opinion, although this reviewer has never worked on GTCs. 

The authors may like to add a short paragraph on the contributing factors in enhancing GTCs efficacy by combining or quoting studies that have demonstrated use of low-fat, low-carb diet, Mediterranean diet and GTCs in the preventive effect on PCa. Demonstrating potential co-benefits in the prevention of other cancers, Colorectal cancers, for example would also enhance the review. 
